# Diagnosis and prognosis of serum Fut8 for epilepsy and refractory epilepsy in children

**Yunxiu Huang[1], Zhou Zhang[2], Linmu Chen[3]***

**1** Department of Laboratory Medicine, Zhongshan People's Hospital, Zhongshan, Guangdong Province, China, **2** Department of Pharmacy, Shenzhen Children's Hospital, Shenzhen, Guangdong Province, China, **3** Department of Pharmacy, Zhongshan People's Hospital, Zhongshan, Guangdong Province, China

\* chenlinmu@gmail.com

**Data Availability Statement:** All relevant data are within the paper and its Supporting Information files.

**Funding:** This study was supported by the Zhongshan Social Welfare and Foundation Special

## Abstract

With adequate serum concentration of antiepileptic drugs, the epilepsy symptoms in many patients still cannot be controlled well. The alteration of glycosyltransferase has obvious influence on the pathogenesis of epilepsy. In this study, we focus on the diagnostic and prognostic value of fucosyltransferase 8 (Fut8) on epilepsy and refractory epilepsy. Serum samples of 199 patients with epilepsy, 59 patients with refractory epilepsy and 22 healthy controls who were diagnosed in Shenzhen Children's hospital from August 2018 to August 2019 were collected. The level of lectins was further analyzed by lectin chip and enzyme linked immunosorbent assay (ELISA). The diagnostic value of serum Fut8 for epilepsy and refractory epilepsy was evaluated by receiver operating characteristic curve. Finally, the difference in the recurrence rate of convulsion in patients with epilepsy or refractory epilepsy within 2 years were observed in different Fut8 expression patients. The concentration of valproic acid (VPA) were significant different between epilepsy and refractory epilepsy group. The expression of α1, 6-fucosylation and Fut8 was significantly increased in the refractory epilepsy group compared with healthy controls. The area under the curve of Fut8 as a biomarker for predicting epilepsy or refractory epilepsy was 0.620 and 0.856, respectively. There was a significant difference in the recurrence rate of convulsion within 2 years in the children with refractory epilepsy ($p = 0.0493$) not epilepsy ($p = 0.1865$) between the high and low Fut8 expression groups. Fut8 was one of the effective indicators for the diagnosis and prognosis of refractory epilepsy.

## Introduction

Epilepsy is one of the most common clinical neurological diseases, which is chronic, repetitive, and spontaneous [1]. It is characterized by excessive synchronization and abnormal discharge of brain neurons [2]. The incidence rate of epilepsy is about 0.41‰ to 1.87‰ in children, which bring serious economic and mental burdens to families and society [3]. At present, the main clinical treatment methods of epilepsy include antiseizure medications (ASMs), surgery, ketogenic diet and neuron stimulation [4]. After treatment with ASMs, the symptom of epilepsy in about 80% of patients could be controlled effectively [5]. Nevertheless, about 20% of

Project (grant number: 2020B3004).The funders had no role in study design, data collection and analysis, decision to publish, or preparation of the manuscript.

**Competing interests:** The authors have declared that no competing interests exist.

epileptic patients, as defined as refractory epilepsy, fail to respond to ASMs [6]. Refractory epilepsy often means a worse prognosis, longer treatment time, and a greater financial burden on the family. VPA, which has been used for more than 30 years, is effective in the treatment of many different types of focal and generalized epileptic seizure [7]. In clinical work, drug concentration monitoring technology is used to better understand the causes of adverse events and poor efficacy. However, the serum concentration of VPA reaches the treatment interval concentration (50–100μg/ml) [8], but the epilepsy symptoms in patients with refractory epilepsy still cannot be controlled well. Therefore, it is particularly important to find an effective predictor to identify patients with refractory epilepsy as early as possible.

Protein glycosylation is the process of attaching sugar chains to proteins under the control of glycosyltransferases [9]. Due to the wide distribution and multiple functions of glycoproteins, incorrect synthesis of sugar chains can lead to abnormalities in multiple systems, such as defects in the development of the nervous system, mental retardation, malformations and epilepsy [10]. This wide range of characteristics reflects the key role of glycoproteins in embryonic development, differentiation and maintenance of cell function [11]. Using in vitro and in vivo models of epileptiform activity, Stewart et al found that acutely increasing O-linked beta-N-acetylglucosamine (O-GlcNAc) levels can significantly attenuate ongoing epileptiform activity and prophylactically dampen subsequent seizure activity [12]. Cuan et al found that the compound stiripentol with good antiepileptic activity can increase the fluorescence intensity of pehaiut agglutinin lectin and vicia villosa lectin [13]. Further work confirmed that the glycosylation modification of asparagine residues of various ion channels such as Cav3.2 T-type calcium ion channel protein will affect its permeability and other functions [14]. The results of our preliminary experiments also showed that the level of N-glycosylation modification in the brain of refractory epilepsy rats was significantly higher than that in the epilepsy group, suggesting that the occurrence of refractory epilepsy is related to the changes of glycosylation modification.

The GDP-fucose was transferred to acetylglucosamine which is connected to asparagine in the form of alpha-1,6 glucoside bond. It resulted in alpha-1,6 fucosyltransferase modification (also known as core fucosyltransferase modification) catalyzed by fucosyltransferase 8(Fut8) [15]. Fut8 is the only glycosyltransferase that catalyzes the fucose modification of protein core in mammals, and plays an important regulatory role in the normal physiological function of glycoproteins [16]. Previous studies have confirmed that Fut8 is indeed closely related to the occurrence of epilepsy. Eight individuals with pathogenic variants in FUT8 (NM_178155.2) have been reported by Bobby et al. All were reported to have epilepsy and microcephaly, no other neurological or immunological findings were available [17,18]. Our previous lectin microarray experiment showed that serum LCA (Lens Culinaris Agglutinin, means α1, 6-fucosylation) levels was higher in patients with epilepsy and refractory epilepsy. Considering the important role of alpha-1,6 fucosylation, we speculated that fut8 might be an important factor for epilepsy disease.

By detecting the expression of Fut8 in serum of patients with epilepsy or refractory epilepsy, we clarified the value of Fut8 as a diagnostic and prognostic factor for epilepsy and refractory epilepsy, hoping to provide a basis for the early detection and treatment of refractory epilepsy.

## Experimental procedures

### Collection of clinical serum samples

Serum samples of 199 epilepsy patients (Epilepsy group), 59 refractory epilepsy patients (Refractory Epilepsy group) and 22 healthy controls (Control group) in Shenzhen Children's hospital were collected from August 2018 to August 2019. All enrolled patients took oral

sodium valproate solution with twice a day (excluding other dosage forms such as tablets, sustained-release tablets and injections) alone or with other ASMs. Inclusion criteria: (1) In line with the diagnostic criteria for epilepsy in Clinical Diagnosis and Treatment Guidelines: Volume of Epilepsy (2015 Revised Edition). Refractory epilepsy is that the seizures are still not completely controlled after sufficient amount and sufficient course of reasonable treatment of two or more ASMs; (2) 1 month to 12 years old; (3) complete clinical data. Exclusion criteria: (1) secondary epilepsy;(2) with serious organic diseases; (3) serious congenital metabolic abnormalities; (4) complicated with other diseases of the brain and nervous system. The basic information of the above patients, such as age, sex, body weight, dosage and other parameters were collected through hospital information system. Patients were followed up from 1 August 2018 to 1 August 2020. This study was conducted in accordance with the Declaration of Helsinki. The studies involving human participants were reviewed and approved by the Ethics Board of Shenzhen Children's hospital. Written informed consent to participate in this study was provided by the participants' legal guardian/next of kin.

## Detection of VPA concentration

After taking the medicine to a steady state, blood is collected before the next administration. A total 1.5–2 ml of venous blood sample were placed in a centrifuge with centrifuge at 4000 r/min for 5 min. Total 100μl of serum was used for detection in a biochemical analyzer. At present, it is believed that the serum concentration of VPA in the range of 50–100μg/ml can effectively control clinical symptoms of epilepsy [8].

## Lectin chip

After obtaining the serum sample from patients, the samples were sent to the company to screen for multiple lectin changes by lectin chip.

## ELISA

The procedure of ELISA is according to the instructions of the reagent supplier. The main steps are as follows: set up 8 standard wells, add 100μl of sample diluent to each well, add 100μl of standard substance to the first well, and dilute sequentially after mixing to ensure that the final volume of each well is 100μl, and the eighth well is a blank control. Add 100 μl of the sample to be tested to each well of the test product; place the reaction plate at 37˚C for 120 minutes, and wash it thoroughly for 4–6 times, and then print to dry. Add 50μl of antibody working solution to each well, mix well and put it at 37˚C for 60 minutes. After washing the plate, add 100 μl of enzyme-labeled antibody working solution to each well, and place the reaction plate at 37˚C for 60 minutes. After washing the plate, add 100μl of substrate solution to each well, and let it react in the dark at 37˚C for 5–10 minutes; add 50μl of stop solution to each well to mix, and measure the absorbance at 450 nm.

## Statistics and analysis

GraphPad 8.0 and SPSS 26.0 software were used for statistical analysis of data. The Chi-square test was used for non-parametric data, and mean ± standard deviation (SD) was used for descriptive continuous variables. The independent samples t-test was used for data analysis if the data obeys a normal distribution. If the data did not obey the normal distribution, the Mann–Whitney U-test was used for data analysis. The Kaplan-Meier method was used for the analysis of survival curves, and log-rank method was used for inspection. A p-value$<$0.05 indicated the difference was statistically significant.

**Table 1. Characteristics of the control, epilepsy and refractory epilepsy patients.**

|  | Control group (n = 22) | Epilepsy group (n = 199) | Refractory epilepsy (n = 59) | p |
|---|---|---|---|---|
| Age (months) | 50.7±9.7 | 49.3±2.3 | 31.1±2.9 | 0.0003 |
| Gender (F/M) | 13/9 | 117/82 | 36/23 | 0.9543 |
| Course of disease (months) | / | 7.5±1.6 | 14.3±3.9 | <0.0001 |
| Seizure type (Generalized/Focal) | / | 60/139 | 28/31 | 0.0186 |
| Drugs (mono/polytherapy) | / | 48/151 | 1/58 | <0.0001 |
| Family history(Y/N) | / | 10/189 | 8/51 | 0.0377 |

## Results

### Basic information of clinical patients

The clinical characteristics of 199 Epilepsy patients, 59 Refractory epilepsy patients and 22 healthy controls participated in this study were shown in Table 1.

### Comparisons of clinical characteristics and serum VPA concentration in the epilepsy and refractory epilepsy patients

As shown in Fig 1A, the weight of patients in the refractory epilepsy group (13.2±4.9 kg) was significantly lower than that in the control group (17.2±2.2 kg). But the weight of patients between the epilepsy group (16.4±6.6 kg) and control group (17.2±2.2 kg) has no difference. The dosage of patients in the refractory epilepsy group (11.6±3.7 mg/kg once) was significantly higher than that in the epilepsy group (9.5±2.6 mg/kg once) (Fig 1B). The serum concentration of VPA did not change significantly between the epilepsy group (57.2±18.6 μg/ml) and refractory epilepsy group (54.9±20.4 μg/ml) (Fig 1C).

### Changes in the expression of different lectins in the patient's serum

As shown in Fig 2A, the expression of AAL (total fucosylation) and LCA (α1, 6-fucosylation) had significant increase in the refractory epilepsy group. Through ELISA assay, we confirm

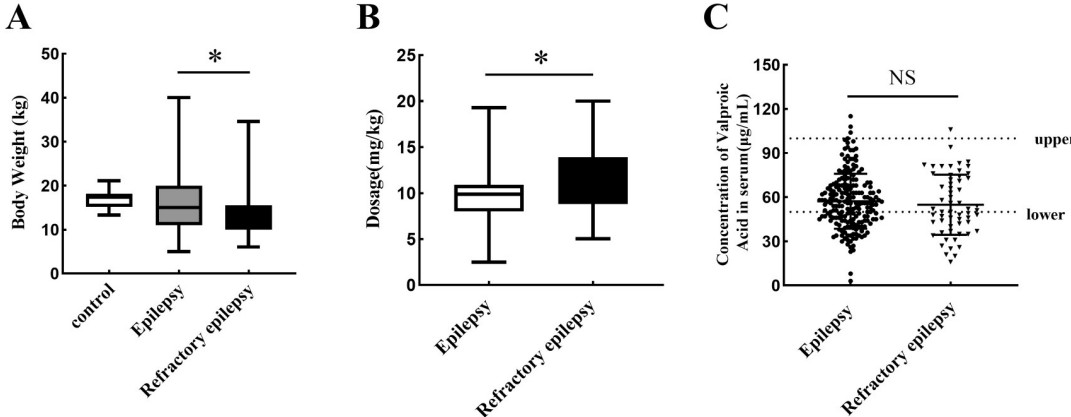

**Fig 1. General information and serum VPA concentration in the epilepsy and refractory epilepsy patients. A**. The difference of weight between the three groups. **B**. The difference of sodium valproate dosage between the two groups. **C**. Changes of serum VPA concentration, upper shows 100 μg/ml, lower shows 50 μg/ml. NS shows no difference, * indicated p <0.05.

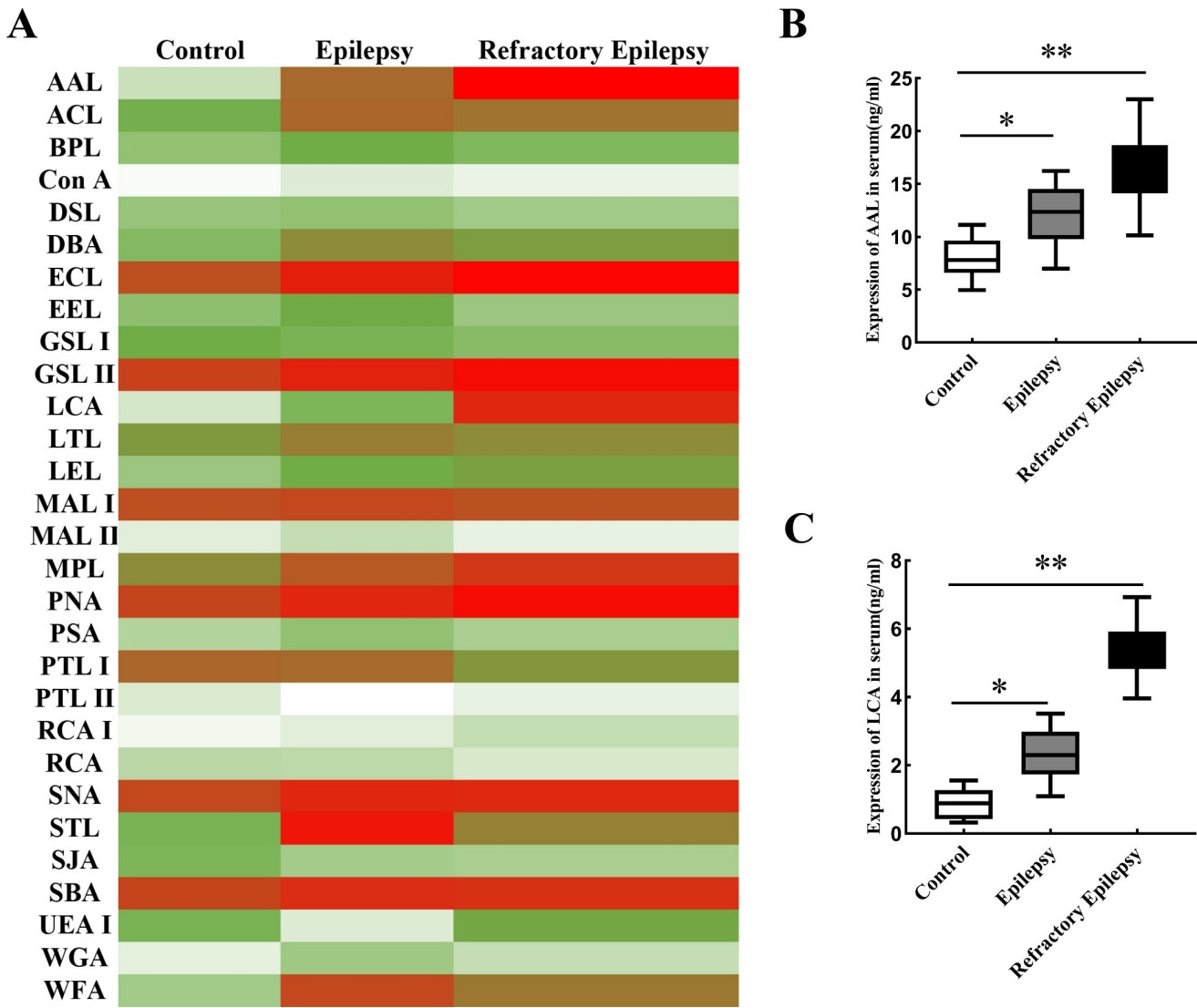

**Fig 2. Changes in the expression of lectins in the patient's serum. A**. Changes of lectins were examined by lectin chip. **B**. ELISA was used to detect AAL expression. **C**. ELISA was used to detect LCA expression. * indicated *p* <0.05, ** indicated *p* <0.01.

the changes of AAL (Fig 2B) and LCA (Fig 2C). The LCA lectin represents the glycosylation of α1, 6-fucosylation, which is catalyzed by Fut8.

## Correlation between serum Fut8 expression and valproic acid concentration

In clinical work, concentration of VPA is often detected with blood drug concentration monitoring technology to better understand the causes of adverse events and poor efficacy. To clarify the correlation between serum Fut8 expression and valproic acid concentration, we detected the level of Fut8 and LCA in the serum of patients by ELISA. As shown in Fig 3A, we found that the expression of Fut8 was significant upregulated in the refractory epilepsy group (95.8±38.1 ng/ml) compared with the control group (54.2±13.1 ng/ml). However, the expression of Fut8 in the epilepsy group (63.7±23.4 ng/ml) has a increase trend compared with the control group without significant difference. The results of Pearson Correlation analysis

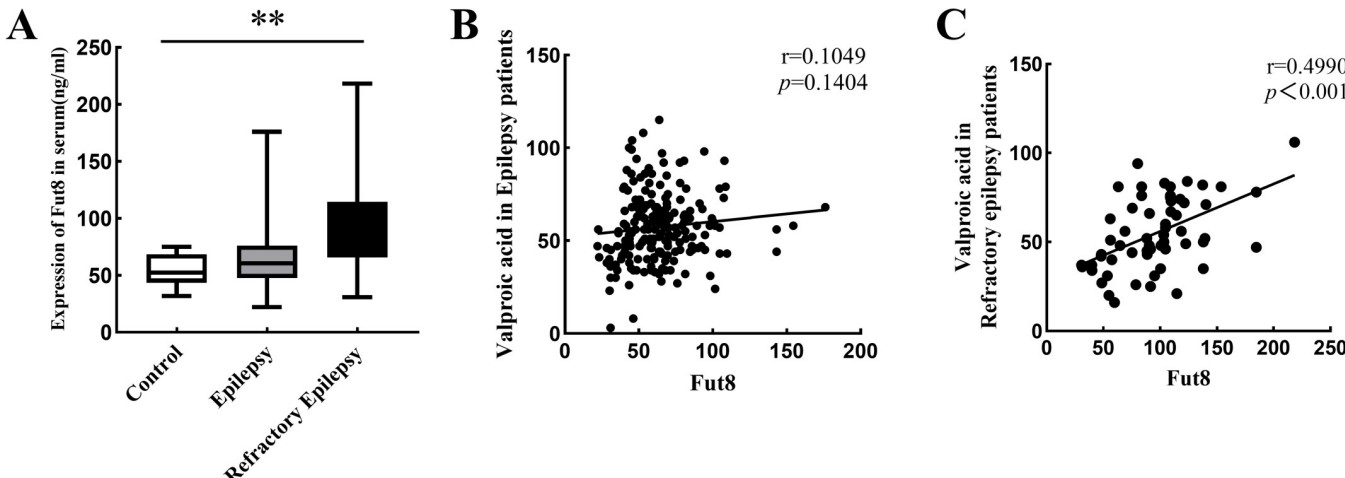

**Fig 3. Correlation between serum Fut8 expression and valproic acid concentration. A**. The expression of Fut8 in three groups were detected by ELISA. **B**. Pearson Correlation analyzed the correlation between Fut8 and valproic acid in epilepsy patients. **C**. Pearson Correlation analyzed the correlation between Fut8 and valproic acid in refractory epilepsy patients. * indicated $p<0.05$, ** indicated $p<0.01$.

showed that the correlation between Fut8 and VPA was not significant (r = 0.1049, p = 0.1404) in the epilepsy patients (Fig 3B), but it is significant (r = 0.4990, p<0.001) in the refractory epilepsy patients (Fig 3C).

## Diagnostic value of Fut8 for epilepsy and refractory epilepsy

We evaluated the diagnostic value of serum Fut8 levels for epilepsy or refractory epilepsy. A receiver operating characteristic curve showed that the expression of Fut8 could distinguish epilepsy patients from the healthy controls, and the area under curve (AUC)was 0.620. Moreover, the sensitivity was 25.5%, and the specificity was 95.5% (Fig 4A). The expression of Fut8

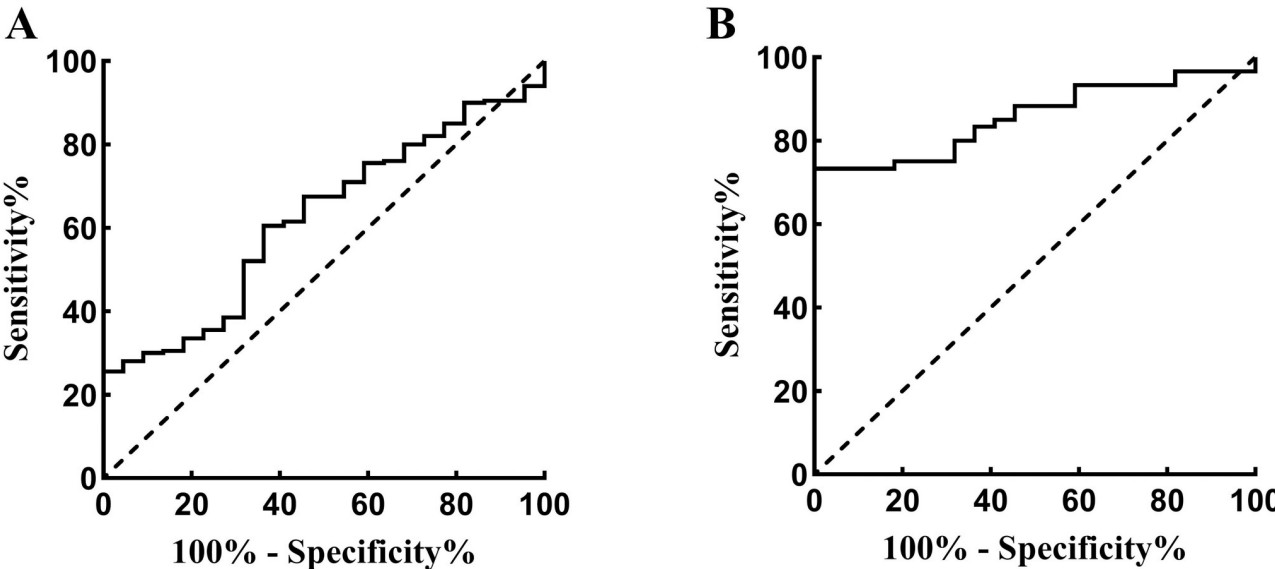

**Fig 4. Receiver operating characteristic curve for evaluating epilepsy and refractory epilepsy by Fut8. A**. Fut8 could distinguish the epilepsy patients from healthy controls with an AUC of 0.620. **B.** Fut8 could distinguish the refractory epilepsy patients from healthy controls with an AUC of 0.856.

**Table 2. Evaluating epilepsy and refractory epilepsy by Fut8.**

|  | AUC | *p* | Sensitivity (%) | Specificity (%) | Youden index |
|---|---|---|---|---|---|
| Fut8 for epilepsy | 0.620 | 0.064 | 25.5 | 95.5 | 0.236 |
| Fut8 for refractory epilepsy | 0.856 | <0.0001 | 73.3 | 90.9 | 0.688 |

could distinguish refractory epilepsy patients from the healthy controls, and the AUC was 0.856. Moreover, the sensitivity was 73.3%, and the specificity was 90.9% (Fig 4B). The detailed information between the two groups was displayed in Table 2.

## Correlations between the serum Fut8 with clinical characteristics in epilepsy group or refractory epilepsy group

To investigate whether Fut8 was related to the clinical parameters, epilepsy or refractory epilepsy patients were grouped into a relatively low/high level serum Fut8, according to the median. Detailed information about the clinical characteristics of the epilepsy or refractory epilepsy patients was presented in Table 3, including gender, age, body weight, Concentration of VPA in serum et al. These data suggested that expression of Fut8 in serum was significantly related to body weight (p = 0.028) in epilepsy group. Also, we found that expression of Fut8 in serum was significantly related to the concentration of VPA in serum (p = 0.034) and seizure of family history (p = 0.026) in refractory epilepsy group.

## Recurrence of convulsion in patients with epilepsy and refractory epilepsy within 2 years

According to survival analysis, the incidence of epileptic seizures in epilepsy patients within 2 years between high and low Fut8 expression were not significantly difference (p = 0.1865, Fig 5A). Further, high expression of Fut8 in serums of refractory epilepsy patients was a significant correlated with the incidence of epileptic seizures within 2 years (p<0.05, Fig 5B).

**Table 3. Correlations between the serum Fut8 with clinical characteristics in epilepsy and refractory epilepsy group.**

| Characteristics | Total | Level of Fut8 in epilepsy | | χ2 value | *p* | Total | Level of Fut8 in refractory epilepsy | | χ2 value | *p* |
|---|---|---|---|---|---|---|---|---|---|---|
|  |  | Low, n = 111 | High, n = 88 |  |  |  | Low, n = 30 | High, n = 29 |  |  |
| Gender |  |  |  |  |  |  |  |  |  |  |
| Male | 117 | 66 | 51 | 0.046 | 0.830 | 36 | 18 | 18 | 0.027 | 0.871 |
| Female | 82 | 45 | 37 |  |  | 23 | 12 | 11 |  |  |
| Age |  | 47.4±9.1 | 51.8±7.7 |  | 0.368 |  | 30.0±7.0 | 32.3±8.1 |  | 0.703 |
| Body weight |  | 15.7±5.6 | 18.5±10.9 |  | 0.028 |  | 13.3±4.0 | 13.2±5.8 |  | 0.924 |
| Concentration of VPA in serum |  | 57.5±15.3 | 56.9±17.9 |  | 0.829 |  | 51.0±14.6 | 58.8±11.7 |  | 0.034 |
| Seizure of Family history |  |  |  |  |  |  |  |  |  |  |
| Yes | 10 | 7 | 3 | 0.863 | 0.352 | 8 | 7 | 1 | 4.975 | 0.026 |
| No | 189 | 104 | 85 |  |  | 51 | 23 | 28 |  |  |
| Seizure type |  |  |  |  |  |  |  |  |  |  |
| Generalized | 60 | 32 | 28 | 0.208 | 0.648 | 28 | 16 | 12 | 1.416 | 0.234 |
| Focal | 139 | 79 | 60 |  |  | 31 | 14 | 17 |  |  |
| Drugs |  |  |  |  |  |  |  |  |  |  |
| monotherapy | 48 | 27 | 21 | 0.006 | 0.939 | 1 | 1 | 0 | 0.983 | 0.321 |
| polytherapy | 151 | 84 | 67 |  |  | 58 | 29 | 29 |  |  |

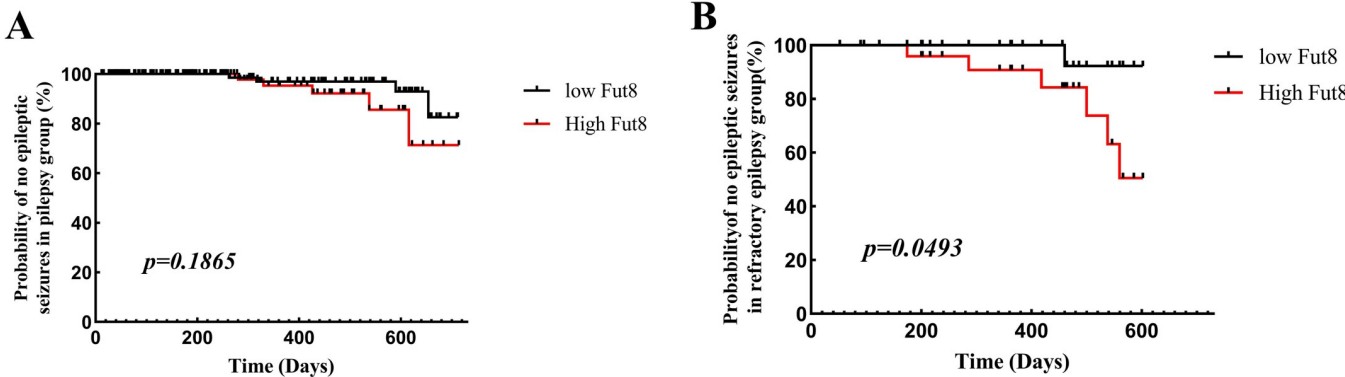

**Fig 5. Kaplan–Meier curve illustrating incidence of epileptic seizures of patients during the follow-up period, based on Fut8 expression. A**. The incidence of epileptic seizures in epilepsy patients within 2 years between high and low Fut8 expression. **B**. The incidence of epileptic seizures in refractory epilepsy patients within 2 years between high and low Fut8 expression.

## Discussion

Sodium valproate, which is the most widely used broad-spectrum antiepileptic drug in clinical, has a rapid and good effective treatment for systemic seizure [7]. In this study, we found that the serum concentration of VPA was not different between the epilepsy group and refractory epilepsy group. And serum VPA concentration in more than half (52.5%) refractory epilepsy patients was between 50–100 µg/ml. It indicated that the poor control of epilepsy symptoms in patients with refractory epilepsy cannot be attributed to the lack of serum concentration of antiepileptic drugs alone, and blindly increasing the dose of administration is not the optimal method for the treatment of refractory epilepsy. Previous studies have confirmed that when the concentration of VPA is higher than 100 µg/ml, the incidence of adverse reactions increases significantly [19]. The blood concentration of sodium valproate is affected by various factors in patients with epilepsy, such as age, body weight, family history of epilepsy, and other case states [20]. Therefore, the monitoring of the individual's blood concentration is still of great significance for predicting the occurrence of adverse reactions not the effectiveness.

Differences in initial clinical features among the three groups were observed. The inconsistency of the above basic data between the two groups were mainly related to the actual clinical situation. Because of the poor control of the epilepsy symptoms in the refractory epilepsy group, clinicians are more inclined to increase the dosage and use multiple drugs for treatment. At the same time, refractory epilepsy is mostly related to the syndrome caused by genetic abnormalities. Gene mutations often affect the growth and development of patients. Therefore, clinical features of young children with dysplasia are more common.

The glycosylation has been confirmed to have an important influence on the occurrence and development of epilepsy [21,22]. Our results showed that the expression levels of Fut8 protein and the α1, 6-fucosylation levels in the refractory epilepsy group were significantly upregulated. However, the factors influencing the up-regulation of Fut8 were still unknown, such as initial patient clinical characteristics and ASMs. Based on the detection of Fut8 expression, we were divided into high and low expression group of Fut8 according to its mean value. Experimental results show that expression of Fut8 in serum was significantly related to the seizure of family history in refractory epilepsy group (p = 0.026). This is consistent with previous reports that Fut8 gene mutation can cause brain developmental disorders [23]. It is possible that partial refractory epilepsy caused by gene defects may be related to Fut8 gene mutation. The relationship between serum VPA and Fut8 level can be confirmed from many aspects.

Experimental results show that expression of Fut8 in serum was significantly related to the concentration of VPA in serum (p = 0.034) in refractory epilepsy group. The results of person correlation analysis showed that the correlation between Fut8 and VPA is significant. It suggests that the high serum VPA concentration in patients with refractory epilepsy may be closely related to the high expression of Fut8. VPA may be one of the factors affecting Fut8, considering the increasing number of medications in children with refractory epilepsy compared with children with epilepsy, and the changing trend of Fut8 among the three groups. However, it is controversial whether other ASMs affect the expression of Fut8 as well. According to a review of the literature, certain proteins or peptides appear to be affected by antiepileptic drugs. However, the effect depends on the kind of drugs. Costa et al found that ghrelin was unchanged by ASMs, including the first (e.g., carbamazepine), second (levetiracetam), and third (lacosamide) generation of anticonvulsants [24]. But Gungor et al found that the weight gain in using valproic acid may be associated with the increase in ghrelin level in the early treatment period [25]. Therefore, we will focus on the changes of Fut8 before and after administration in the future experiment to explore the important role of Fut8 in the pathogenesis of epilepsy. Therefore, serum of children with epilepsy or refractory epilepsy before and after treatment should be collected and Fut8 expression level should be detected to answer the above questions.

Glycosylation which is a fundamental cellular process catalyzed by glycosyltransferase occurs in the lumen of both the Golgi apparatus and the endoplasmic reticulum in eukaryotes [26–28]. So, what is the target protein that may play a role in this process of epilepsy? Current theories on epilepsy resistance include the drug transporter hypothesis, drug target hypothesis, and epilepsy inherent severity hypothesis [29–31], commonly used ASMs are the substrate drugs of P-glycoprotein (P-gp) [32–34], which is the most widely studied multidrug transporter [35]. Glycosylation of P-gp occurs on the first extracellular loop, which contains three putative glycosylation sites [36]. Glycosylation of P-gp was shown to be important for proper quality control of P-gp in the endoplasmic reticulum [37] and proper transport of P-gp to the plasma membrane [36]. Previous literature suggested that P-pg is highly expressed in patients with refractory epilepsy [38]. Therefore, we speculated that the high expression of Fut8 promoted the upregulation of P-pg glycosylation level, leading to more antiepileptic drugs being transported out of the brain, resulting in insufficient drug concentration at the seizure site in the brain. It may be one of the reasons why the actual effect was not good even though the serum VPA concentration reached the requirement. In subsequent experiments, we will detect the glycosylation modification changes of P-pg in brain tissues, and use transgenic mice to construct epileptic models to observe the effects of P-pg glycosylation knockdown on epileptic drug distribution and anti-epileptic effect.

In conclusion, our experiment confirmed that Fut8 is one of the early diagnoses for predicting refractory epilepsy, but Fut8 was be affected by demographic and clinical features such as the concentration of VPA and seizure of family history.

## Supporting information

**S1 File.**
(XLSX)

## Acknowledgments

We would like to thank all the patients and controls that participated in this study.

## Author Contributions

**Data curation:** Yunxiu Huang, Linmu Chen.

**Formal analysis:** Yunxiu Huang, Linmu Chen.

**Funding acquisition:** Linmu Chen.

**Investigation:** Yunxiu Huang, Zhou Zhang.

**Resources:** Zhou Zhang.

**Writing – original draft:** Yunxiu Huang, Linmu Chen.

**Writing – review & editing:** Yunxiu Huang, Linmu Chen.

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
