## [Decision Letter · Decision Letter 0]

15 Feb 2023

PONE-D-23-00619Increased expression of Fut8 is an appropriate predictor for the diagnosis and prognosis of refractory epilepsy not epilepsyPLOS ONE

Dear Dr. Chen,

Thank you for submitting your manuscript to PLOS ONE. After careful consideration, we feel that it has merit but does not fully meet PLOS ONE’s publication criteria as it currently stands. Therefore, we invite you to submit a revised version of the manuscript that addresses the points raised during the review process. In addition to the reviewers' comments, the title of your manuscript should be more consistent with the reported findings, which appear to be specific for valproate.

We look forward to receiving your revised manuscript.

Kind regards,

Giuseppe Biagini, MD

Academic Editor

PLOS ONE

Journal Requirements:

"This study was supported by the Zhongshan Social Welfare and Foundation Special Project (grant number: K2020B3004)."

"YES - Specify the role(s) played."

Reviewers' comments:

Reviewer's Responses to Questions

**Comments to the Author**

1. Is the manuscript technically sound, and do the data support the conclusions?

Reviewer #1: Partly

2. Has the statistical analysis been performed appropriately and rigorously? 

Reviewer #1: Yes

3. Have the authors made all data underlying the findings in their manuscript fully available?

Reviewer #1: Yes

4. Is the manuscript presented in an intelligible fashion and written in standard English?

Reviewer #1: Yes

5. Review Comments to the Author

Reviewer #1: The article by Chen et al is of interest. However, it should be extensively revised before publication.

1. The groups should be renamed. Indeed, the 199 patients with epilepsy and 59 patients with refractory epilepsy are all affected by epilepsy. The difference is that one group should be drug-sensible and the other drug-resistant.

2. In the abstract it is mentioned that “the alteration of glycosyltransferase has obvious influence on the pathogenesis of epilepsy. In this study, we focus on the diagnostic and prognostic value of Fucosyltransferase 8 (Fut8) on epilepsy or refractory epilepsy.” However, in the main text, it is not clear enough why the authors decided to focus especially on Fut8.

3. Lines 28-29: “The area under the curve of Fut8 as a biomarker for predicting epilepsy or refractory epilepsy was 0.620 and 0.856, respectively.” Is the area under the curve significant different between groups?

4. The new proposed term for medications aimed to treat epilepsy is antiseizure medications (ASMs). This term replaces the previous one (anti-epileptic drugs, AEDs). Please, fix this issue.

5. As Fut8 is proposed to be an effective indicator for the diagnosis and prognosis of refractory epilepsy, the introduction should be centered on the physiological role of Fut8 and why it could be important in the diagnosis and prognosis of epilepsy, especially refractory epilepsy. The introduction should be rearranged, as some information is missing, whereas other information is just introduced in the second half of the introduction.

6. The meaning of the abbreviations (e.g., O-GlcNAc, STP, etc.) should be added.

7. It is not clear when the collection of serum samples was performed. For instance, it not clear whether the patients were fasting, or not, or how many days, months, or years patients took the therapy before being included in the study and being tested for Fut8 expression.

8. A table with the percentage of patients per group, using a specific ASM for both mono and polytherapy, should be added.

9. The characteristics of controls should be also listed.

10. Is there any correlation between serum Fut8 expression and age, body weight and type of seizures of the patients?

11. Are the serum levels of Fut8 changed by ASMs? In other words, if you divide patients in relation to the treatment (e.g., VPA, VPA plus other ASMs…) and not in relation to the drug response (drug-sensible vs drug-resistant), do you find any significant differences in the levels of Fut8?

12. Lines 185-186: “Glycosyltransferase, which add sugar chains to the downstream target protein mainly in the endoplasmic reticulum and Golgi.” Please, check!

13. The discussion should definitely be improved.

14. The conclusion drawn by the authors (“Fut8 is one of the early diagnoses for predicting refractory epilepsy, and the elevation of serum Fut8 is conducive to the early diagnosis and treatment of refractory epilepsy”) is negatively affected by the data collected in the experiment. Indeed, as shown in Table 1, the age and seizure type are significant different between groups. Moreover, the body weight is also significantly different between groups, as demonstrated in Figure 1A. For these reasons, it should be concluded that the serum levels of Fut8 did not depend only on the response to treatment, but also on demographic and clinical features such as the type of epilepsy, age, and body weight.

15. In this regard, the similarities with other published papers (Costa et al., 2022 doi.org/10.3390/jpm12040527) could be discussed. Indeed, it was reported that total ghrelin plasma levels and the ghrelin-to-DAG ratio were determined by demographic and clinical features such as the type of epilepsy, age, head circumference, and BMI.

16. For the above-mentioned reasons, the title should be changed, and the article extensively revised.

6. PLOS authors have the option to publish the peer review history of their article (what does this mean?). If published, this will include your full peer review and any attached files.

Reviewer #1: No

---

## [Author Response · Author response to Decision Letter 0]

22 Mar 2023

We would like to thank the editor and reviewers for these precious comments concerning my manuscript entitled “Increased expression of Fut8 is an appropriate predictor for the diagnosis and prognosis of refractory epilepsy not epilepsy, PONE-D-23-00619”. These comments are very valuable and helpful for revising and improving my paper, as well as the important guiding significance to my research. We have studied comments carefully and have made corrections which we hope meet with approval. In this revised version, changes to our manuscript within the document were all highlighted by red colored text. Point-by-point responses to the editor and reviewer are as follows: 

Journal Requirements:

R: We have made same changes according to the PLOS ONE's style requirements.

R: We are sorry for the mistake. We have provided the correct grant numbers for the awards in the ‘Funding Information’ section in the revised cover letter.

3. Thank you for stating the following in the Funding Section of your manuscript: "This study was supported by the Zhongshan Social Welfare and Foundation Special Project (grant number: K2020B3004)." We note that you have provided funding information that is not currently declared in your Funding Statement. However, funding information should not appear in the Acknowledgments section or other areas of your manuscript. We will only publish funding information present in the Funding Statement section of the online submission form. Please remove any funding-related text from the manuscript and let us know how you would like to update your Funding Statement. Currently, your Funding Statement reads as follows: "YES - Specify the role(s) played." Please include your amended statements within your cover letter; we will change the online submission form on your behalf.

R: Thanks! The funding-related text has been removed from the manuscript. The statements about the funding information have been included in the revised cover letter.

R: Thanks! The ethics statement appeared in the “Collection of clinical serum samples” section.

5. In addition to the reviewers' comments, the title of your manuscript should be more consistent with the reported findings, which appear to be specific for valproate.

R: Thanks for your good question. 

As a broad-spectrum antiepileptic drug, sodium valproate (VPA) is widely used in children with epilepsy. In our previous clinical work, we found that serum VPA concentration was up to the standard line (50-100 μg/ml) in both epilepsy and refractory epilepsy children, but the curative effect was quite different. Combined with relevant literature, we start from the theory of drug transporter receptor, considering that the function difference of drug transporter receptor may affect the actual concentration of drugs in the brain. Glycosylation modification is an important factor affecting the function of drug transporters. Therefore, we determined the children's subsequent reactivity to drugs by detecting the serum concentration of Fut8.

In the discussion part, we also discussed the influence of the drug itself on Fut8 expression, and made it clear that the follow-up experiment will carry out the detection of Fut8 expression before and after drug treatment.

Based on the above reasons, we think that serum VPA concentration in this experiment as an important clinical parameter for clinical patients might me more suitable, and the purpose of the experiment is still to explore the changes of Fut8 in children with epilepsy and refractory epilepsy. 

Reviewers' comments:

Reviewer's Responses to Questions

Comments to the Author

1. Is the manuscript technically sound, and do the data support the conclusions?

Reviewer #1: Partly

2. Has the statistical analysis been performed appropriately and rigorously?

Reviewer #1: Yes

3. Have the authors made all data underlying the findings in their manuscript fully available?

Reviewer #1: Yes

4. Is the manuscript presented in an intelligible fashion and written in standard English?

Reviewer #1: Yes

 

5. Review Comments to the Author

Reviewer #1: The article by Chen et al is of interest. However, it should be extensively revised before publication.

1) The groups should be renamed. Indeed, the 199 patients with epilepsy and 59 patients with refractory epilepsy are all affected by epilepsy. The difference is that one group should be drug-sensible and the other drug-resistant.

R: Thank you for your good suggestion. When epilepsy patient treated with more than two antiepileptic drugs is still failed to achieve a sustained seizure free, the epilepsy can be considered refractory epilepsy. The most important difference between epilepsy and refractory epilepsy is indeed essentially sensitive to antiepileptic drugs. However, epilepsy or refractory epilepsy is the first diagnosis for these patients clinically. Therefore, the group named epilepsy or refractory epilepsy may be more suitable for the clinical situation. Thanks again for this question.

2) In the abstract it is mentioned that “the alteration of glycosyltransferase has obvious influence on the pathogenesis of epilepsy. In this study, we focus on the diagnostic and prognostic value of Fucosyltransferase 8 (Fut8) on epilepsy or refractory epilepsy.” However, in the main text, it is not clear enough why the authors decided to focus especially on Fut8.

R: We are sorry for the confusion. In the Fig2, we detect the changes in most common lectins using lectin array. The LCA lectin has a significant change between the two groups. The LCA lectin represents the glycosylation of α1, 6-fucosylation, which is catalyzed by Fucosyltransferase 8. This sentence was added in the Results section (Page8 line 142-143)

3) Lines 28-29: “The area under the curve of Fut8 as a biomarker for predicting epilepsy or refractory epilepsy was 0.620 and 0.856, respectively.” Is the area under the curve significant different between groups?

R: Thanks! The detailed information between the two groups was displayed in Table2 (Page23, line 440).

4) The new proposed term for medications aimed to treat epilepsy is antiseizure medications (ASMs). This term replaces the previous one (anti-epileptic drugs, AEDs). Please, fix this issue.

R: Thank you for the suggestion. The AEDs has been replaced by ASMs in the manuscript.

5) As Fut8 is proposed to be an effective indicator for the diagnosis and prognosis of refractory epilepsy, the introduction should be centered on the physiological role of Fut8 and why it could be important in the diagnosis and prognosis of epilepsy, especially refractory epilepsy. The introduction should be rearranged, as some information is missing, whereas other information is just introduced in the second half of the introduction.

R: Thank you for your good suggestion. We have added some content to emphasize the important role of Fut8 in the onset of epilepsy, which will pave the way for our subsequent research. The content about drug transporter receptor was deleted. Thanks again to the reviewer for the improvement of the Introduction section(Page4-5, line63-74).

6) The meaning of the abbreviations (e.g., O-GlcNAc, STP, etc.) should be added.

R: Thanks! The manuscript had been checked. The meaning of the abbreviations had been added.

7) It is not clear when the collection of serum samples was performed. For instance, it not clear whether the patients were fasting, or not, or how many days, months, or years patients took the therapy before being included in the study and being tested for Fut8 expression.

R: Thanks for the question. 

1.The serum samples in control group were collected from physical examination sample of a normal child. 

2. Treatment with valproate alone or in combination with other antiepileptic drugs, the seizure was not found in epilepsy patient. The illness condition of patient was stable. The serum samples in epilepsy group were collected. 

3. Serum would be collected if the patient is diagnosed with refractory epilepsy, including epileptic syndromes or patients who have failed to respond to multidrug therapy. 

Considering the convenience of clinical practice and ethical requirements, we did not conduct dieting and starvation operations on patients. The time required for blood collection is before the next dose.

8) A table with the percentage of patients per group, using a specific ASM for both mono and polytherapy, should be added.

R: Thanks. The information was added in Table 1(Page23, line438).

9) The characteristics of controls should be also listed.

R: Thanks. Several disease-related indicators in Table 1 were not present in the control group. Therefore, only the age and gender in control group were added Table1 (Page23, line438). 

10) Is there any correlation between serum Fut8 expression and age, body weight and type of seizures of the patients?

R: Thanks. Correlations between the serum Fut8 with clinical characteristics in epilepsy and refractory epilepsy group was added in Table3(Page24, line443).

11) Are the serum levels of Fut8 changed by ASMs? In other words, if you divide patients in relation to the treatment (e.g., VPA, VPA plus other ASMs…) and not in relation to the drug response (drug-sensible vs drug-resistant), do you find any significant differences in the levels of Fut8?

R: Thanks for your question. It is an interesting question. According to a review of the literature, certain proteins or peptides appear to be affected by antiepileptic drugs. However, the effect depends on the kind of drugs. Costa et al found that ghrelin was unchanged by ASMs, including the first (e.g., carbamazepine), second (levetiracetam), and third (lacosamide) generation of anticonvulsants (PMID: 35455643). But Gungor et al found that the weight gain in using valproic acid may be associated with the increase in ghrelin level in the early treatment period (PMID: 18174556). Therefore, serum of children with epilepsy or refractory epilepsy before and after treatment should be collected and Fut8 expression level should be detected to answer the above questions. However,

Antiepileptic drugs may be one of the factors affecting Fut8, considering the increasing number of medications in children with refractory epilepsy compared with children with epilepsy, and the changing trend of Fut8 among the three groups. But further experiments are needed to determine.

Thanks for the questions provided by the reviewer, we will conduct such experiments in the future to further understand the role of Fut8 in the occurrence of epilepsy. These have been added to the discussion section.

12) Lines 185-186: “Glycosyltransferase, which add sugar chains to the downstream target protein mainly in the endoplasmic reticulum and Golgi.” Please, check!

R: I am sorry for the grammatical mistake. The sentence has been changed (Page12, line 229-231).

13) The discussion should definitely be improved.

R: Thanks. We made many changes to the discussion section with red color. 

14) The conclusion drawn by the authors (“Fut8 is one of the early diagnoses for predicting refractory epilepsy, and the elevation of serum Fut8 is conducive to the early diagnosis and treatment of refractory epilepsy”) is negatively affected by the data collected in the experiment. Indeed, as shown in Table 1, the age and seizure type are significant different between groups. Moreover, the body weight is also significantly different between groups, as demonstrated in Figure 1A. For these reasons, it should be concluded that the serum levels of Fut8 did not depend only on the response to treatment, but also on demographic and clinical features such as the type of epilepsy, age, and body weight.

R: Thank you for the rigorous expression. Limited to clinical conditions, there are certain differences in some clinical characteristics between the two groups of children. For example: Children with refractory epilepsy due to genetic defects have developmental disorders, leading to the weight loss. The dose of concentration in children with refractory epilepsy would be increased. 

In Table3 (Question 10), correlations between the serum Fut8 with clinical characteristics in epilepsy and refractory epilepsy group was described. Also, we changed the conclusion on the end of Discussion section.

In future, the method of Propensity Score Matching will be used to reduce the confounding effect of confounding factors on results.

15) In this regard, the similarities with other published papers (Costa et al., 2022 doi.org/10.3390/jpm12040527) could be discussed. Indeed, it was reported that total ghrelin plasma levels and the ghrelin-to-DAG ratio were determined by demographic and clinical features such as the type of epilepsy, age, head circumference, and BMI.

R: Thanks. Referring to the above documents, we discussed the influence of some clinical features on Fut8 on the basis of Table3.

16) For the above-mentioned reasons, the title should be changed, and the article extensively revised.

R: Thanks. We have revised the article according to the expert's opinion

Thanks again for the editor and reviewers’ valuable and helpful suggestion for revising and improving my paper.

---

## [Decision Letter · Decision Letter 1]

28 Mar 2023

Diagnosis and prognosis of serum Fut8 for epilepsy and refractory epilepsy in children

PONE-D-23-00619R1

Dear Dr. Chen,

We’re pleased to inform you that your manuscript has been judged scientifically suitable for publication and will be formally accepted for publication once it meets all outstanding technical requirements.

Kind regards,

Giuseppe Biagini, MD

Academic Editor

PLOS ONE

Additional Editor Comments (optional):

Reviewers' comments:

Reviewer's Responses to Questions

**Comments to the Author**

1. If the authors have adequately addressed your comments raised in a previous round of review and you feel that this manuscript is now acceptable for publication, you may indicate that here to bypass the “Comments to the Author” section, enter your conflict of interest statement in the “Confidential to Editor” section, and submit your "Accept" recommendation.

Reviewer #1: All comments have been addressed

2. Is the manuscript technically sound, and do the data support the conclusions?

Reviewer #1: (No Response)

3. Has the statistical analysis been performed appropriately and rigorously? 

Reviewer #1: (No Response)

4. Have the authors made all data underlying the findings in their manuscript fully available?

Reviewer #1: (No Response)

5. Is the manuscript presented in an intelligible fashion and written in standard English?

Reviewer #1: (No Response)

6. Review Comments to the Author

Reviewer #1: All comments have been adressed.

7. PLOS authors have the option to publish the peer review history of their article (what does this mean?). If published, this will include your full peer review and any attached files.

Reviewer #1: No

---

## [Editor Report · Acceptance letter]

5 Apr 2023

PONE-D-23-00619R1 

Diagnosis and prognosis of serum Fut8 for epilepsy and refractory epilepsy in children 

Dear Dr. Chen:

I'm pleased to inform you that your manuscript has been deemed suitable for publication in PLOS ONE. Congratulations! Your manuscript is now with our production department. 

Kind regards, 

on behalf of

Dr. Giuseppe Biagini 

Academic Editor

PLOS ONE